# Exploring the Immunoresponse in Bladder Cancer Immunotherapy

**DOI:** 10.3390/cells13231937

**Published:** 2024-11-22

**Authors:** Inmaculada Ruiz-Lorente, Lourdes Gimeno, Alicia López-Abad, Pedro López Cubillana, Tomás Fernández Aparicio, Lucas Jesús Asensio Egea, Juan Moreno Avilés, Gloria Doñate Iñiguez, Pablo Luis Guzmán Martínez-Valls, Gerardo Server, José Félix Escudero-Bregante, Belén Ferri, José Antonio Campillo, Eduardo Pons-Fuster, María Dolores Martínez Hernández, María Victoria Martínez-Sánchez, Diana Ceballos, Alfredo Minguela

**Affiliations:** 1Immunology Service, Virgen de la Arrixaca University Clinical Hospital (HCUVA), Biomedical Research Institute of Murcia (IMIB), 30120 Murcia, Spain; irl_98@hotmail.com (I.R.-L.); lgarias@um.es (L.G.); josea.campillo@carm.es (J.A.C.); lola.mtz.hdz@gmail.com (M.D.M.H.); vickyms7@hotmail.com (M.V.M.-S.); dceballosf@gmail.com (D.C.); 2Human Anatomy Department, Universidad de Murcia and Campus Mare Nostrum, 30071 Murcia, Spain; eduardo.p.f@um.es; 3Urology Service, Virgen de la Arrixaca University Clinical Hospital (HCUVA), Biomedical Research Institute of Murcia (IMIB), 30120 Murcia, Spain; alicialopezabad@gmail.com (A.L.-A.); pedrolopezcubillana@gmail.com (P.L.C.); gerardoserver@gmail.com (G.S.); josef.escudero@carm.es (J.F.E.-B.); 4Urology Service, Morales Meseguer Hospital, 30008 Murcia, Spain; tomas.fernandez3@carm.es; 5De la Vega Lorenzo Guirao Hospital, 30530 Murcia, Spain; lucasasensioegea@gmail.com; 6Urology Service, Santa Lucia Hospital, 4067 Murcia, Spain; juan.moreno2@carm.es; 7Urology Service, Los Arcos Hospital, 30739 Murcia, Spain; gloria.donate@carm.es; 8Urology Service, Reina Sofía Hospital, 14004 Murcia, Spain; pablol.guzman@carm.es; 9Pathology Service, Virgen de la Arrixaca University Clinical Hospital (HCUVA), Biomedical Research Institute of Murcia (IMIB), 30120 Murcia, Spain; belenferri@msn.com

**Keywords:** bladder cancer, immune response, immunotherapy, BCG, checkpoints

## Abstract

Bladder cancer (BC) represents a wide spectrum of diseases, ranging from recurrent non-invasive tumors to advanced stages that require intensive treatments. BC accounts for an estimated 500,000 new cases and 200,000 deaths worldwide every year. Understanding the biology of BC has changed how this disease is diagnosed and treated. Bladder cancer is highly immunogenic, involving innate and adaptive components of the immune system. Although little is still known of how immune cells respond to BC, immunotherapy with bacillus *Calmette–Guérin* (BCG) remains the gold standard in high-risk non-muscle invasive BC. For muscle-invasive BC and metastatic stages, immune checkpoint inhibitors targeting CTLA-4, PD-1, and PD-L1 have emerged as potent therapies, enhancing immune surveillance and tumor cell elimination. This review aims to unravel the immune responses involving innate and adaptive immune cells in BC that will contribute to establishing new and promising therapeutic options, while reviewing the immunotherapies currently in use in bladder cancer.

## 1. Introduction

Bladder cancer (BC) is a malignant neoplasm with one of the highest cancer prevalences worldwide, with over 430,000 new cases occurring worldwide every year (75% men) [1]. Tobacco smoking [2] and occupational exposure to aromatic amines or chlorinated hydrocarbons [3] are the main risk factors for BC, accounting for at least 50% and 10% of newly diagnosed cases, respectively. According to the Tumor, Node, Metastasis (TNM) classification (Figure 1A), non-muscle invasive BC (NMIBC), which includes carcinoma in situ (CIS), Ta, and T1 stages, comprises 70% of all BC; while the remaining 30% are classified as T2 to T4 muscle-invasive bladder cancer (MIBC) [4]. Treatment of CIS and high-risk NMIBC is based on the trans-urethral resection of tumor (TURBT) followed by intravesical chemotherapy or immunotherapy (IT) with bacillus *Calmette-Guérin* (BCG). The adjuvant treatment approach is based on the European Association of Urology (EAU) guidelines (https://uroweb.org/guidelines/non-muscle-invasive-bladder-cancer accessed on 19 November 2024), with different therapeutic schemes applied according to the patient’s risk category, and with non-response to BCG implying radical cystectomy (RC) because of the high risk of progression [5,6]. However, MIBC is treated with neoadjuvant cisplatin-based therapy, followed by RC and pelvic lymphadenectomy [6,7]. Patients ineligible for cisplatin are treated with carboplatin, although this treatment is associated with shorter overall survival (OS) [6]. The poor prognosis of MIBC offers an opportunity for immunotherapy to improve outcomes and opens a range of treatment combinations. During the past two decades, several revolutionary immunotherapy approaches have taken center stage in cancer therapy. These include immune checkpoint inhibitors (ICIs), such as programmed cell death protein 1 (PD-1)/PD-L1, cytotoxic T-lymphocyte-associated antigen 4 (CTLA-4) [7,8], and others that mediate natural killer (NK) and T cell dysfunction, such as NKG2A/HLA-E [9] and T cell immunoreceptors with Ig and ITIM domains (TIGIT)/CD155 [10]. All of them will be reviewed in this manuscript in relation to the mechanisms guiding the immune response against BC.

## 2. Bladder Cancer Immune Response

Many studies support the idea that the immune system protects against cancer [10,11]. The reduced tumor recurrence after intravesical BCG therapy is a clear demonstration that immune surveillance also operates in BC [12,13]. The first line of defense against cancer are innate immune effectors such as macrophages, neutrophils, dendritic cells (DCs) and NK cells [14,15], which, by working coordinately with the antigen-specific B and T lymphocytes, might eradicate the tumor and provide long-term protection. Nonetheless, to escape immune surveillance, tumors secrete or promote the secretion of immunosuppressive and anti-apoptotic factors, such as transforming growth factor-β (TGF-β), prostaglandin E2 (PGE2), interleukin (IL)-10, and IL-6 [16,17]. In response to these factors, new immune effectors are recruited from the circulation such as neutrophils, FoxP3^+^ regulatory T cells (Tregs) and myeloid-derived suppressor cells (MDSCs) [14]. These will induce a highly immunosuppressive tumor microenvironment (TME) by promoting the expression of the inhibitory molecule PD-L1, the expansion of tumor-associated macrophages (TAMs), the increase in PGE2 production and an aberrant metabolism of glycosaminoglycans [16]. Although few studies have explored the BC immune landscape, it is necessary to understand the basic principles that govern the immune response in order to harness the power of these biological tools against this cancer.

### 2.1. Innate Immune Effectors

Tumor-derived DNA and other damage-associated molecular patterns (DAMPs) promote DC activation and the production of IFN type-I and IFN-γ. DCs are usual resident cells in a healthy bladder, but largely attracted by the tumorigenic process, which depletes DCs from the blood [14,17]. IFNs stimulate DC expression of the costimulatory molecules CD40, CD80, CD86, and mayor histocompatibility complex (MHC) class-II, promoting their maturation, the ability to present tumor antigens and migratory capabilities. Additionally, DCs exposed to IFN type-I produce high levels of IL-12 and IL-15, stimulating further downstream immune activation of Th1 CD4+ and CD8+ T cells in the TME. Plasmacytoid dendritic cells (pDCs) are specialized DCs that release high levels of IFN type-I in response to antigens and link innate and adaptive immunity, playing a critical role in the initial immune response against tumor cells. However, some studies have found that tumor infiltration of pDCs correlates with poorer outcomes, maybe due to their co-localization with regulatory T cells (Tregs) in the TME, which decreases responsiveness of pDCs. In addition, chronic or prolonged exposure to IFN-γ can impede DC differentiation and homeostasis, instead promoting the upregulation of the immunosuppressive molecules PD-L1 on DCs. Moreover, prolonged exposure to IFN-γ induces an IDO-dependent switch from immunogenic to tolerogenic DCs, which favor the activation of Tregs. In fact, high levels of infiltrating DCs in human BC predict progression to muscle invasion, thus suggesting that DCs may be a significant but unhelpful presence in BC [18]. Notably, GSK-3β inhibitors block IFN-γ-mediated IDO expression, enhancing the activity of DC-based vaccines in vivo [19]. New therapeutic strategies are being addressed, employing anti-CD40 antibodies to increase their APC function and their secretion of IL-12, resulting in suppressed tumor growth in mice [20].

Macrophages are present in healthy human bladders, although they are abundantly recruited to the tumor site at all stages of tumor progression [14]. At initial tumor stages, pro-inflammatory M1 macrophages exhibit tumoricidal functions such as phagocytosis, the release of reactive oxygen species, and the secretion of inflammatory cytokines [14,21,22]. However, these cells have functional plasticity and can evolve into tissue repair anti-inflammatory M2 macrophages [22,23], induced by M2 polarizing cytokines (CCL2, IL-10, and TGF-β) and bone morphogenetic protein 4 (BMP-4) produced by bladder tumors [24,25]. In primary tumors, M2 macrophages can stimulate angiogenesis and enhance tumor cell invasion, motility, and intravasation. During metastasis, macrophages prime the pre-metastatic site and promote tumor cell extravasation, survival, and persistent growth. In addition, M2 macrophages can prevent anti-tumor attacks of NK and T cells during tumor progression and after recovery from chemo- or immuno-therapy [23]. In fact, M2 macrophages do not produce CXCL9 or CXCL10, which could recruit anti-tumor Th1 lymphocytes [26]. TAMs have many characteristics in common with M2 macrophages; in fact, high TAM counts have been correlated with poor survival in BC [26,27].

Neutrophils are absent in healthy bladders but, attracted by cytokines secreted by urothelial tumors (CXCL1, CXCL5, and IL-8), abundantly infiltrate BC where they appear to have a largely immunosuppressive role [28]. In BC, a high-circulating neutrophil-to-lymphocyte ratio [29] and high numbers of tumor-associated neutrophils (TANs) [30] are prognostic biomarkers associated with poorer responses to treatment and recurrence-free and overall patient survival, suggesting a role of TANs in cancer progression. Neutrophils, like macrophages, can polarize to the N2 pro-tumor phenotype induced by TGF-β secreted by BC [31]. These TANs promote tumor cell growth and invasion by remodeling the extracellular matrix and induce the angiogenic switch during early tumor progression, modulating tumor cell biology at later stages [32]. However, BCG therapy triggers the release of cytokines (IL-1β, IL-6, IL-8, granulocyte-macrophage colony-stimulating factor -GM-CSF- and TNF-α), which induce a rapid and abundant infiltration of neutrophils into the bladder with a cytotoxic and anti-tumor phenotype [33].

Myeloid-derived suppressor cells (MDSCs) are immature myeloid cells closely related to monocytes and neutrophils normally absent in a healthy bladder. However, they are released from the bone marrow to the peripheral blood and enrolled into tumors due to the chronic inflammatory cancer milieu, where they negatively correlate with the stage, grade, and prognosis of BC [14]. In the TME, MDSCs produce pro-inflammatory cytokines and immunosuppressive molecules such as Arginase 1, inducible nitric oxide synthases (iNOS) and PD-L1, which directly suppress T cell proliferation, reduce CD8+ T cell infiltration, and promote Treg proliferation [34]. In addition, the recruitment of these cells may represent one of the factors underlying BCG failure in NMIBC [35]. Cisplatin treatment, however, depletes MDSCs from the TME, which could constitute one of the anti-tumor effects of this drug.

NK cells are innate cytotoxic lymphocytes playing essential roles in the first line of defense against cancer [36]. These cells have a powerful cytotoxic activity orchestrated by an intricate network of inhibitory and activating signals [37]. NK cells are not present in a healthy human bladder, and their role in BC is inconclusive [38,39]. Early studies found no defects in the ability of peripheral blood NK cells from patients with BC to degranulate in response to MHC-deficient target cells. However, NK cells isolated from tumor and lymph nodes showed a substantial degranulation defect [33]. In addition, higher NK cell infiltration was associated with higher tumor size in NMIBC patients who relapsed after 2 years [40], suggesting an adverse role for these cells. Nonetheless, other studies showed that higher infiltration of the CD56^bright^ NK cell subset was associated with improved BC patient survival. CD56^bright^ NK cells were functionally active, secreting much higher amounts of cytokines than the CD56^dim^ subset [41]. Our group has recently described that the KIR/HLA-I interaction repertoire, involved in NK cell education and function, offers an immunological risk stratification that complements the TNM classification and supports a decisive role of NK cells in BC immune surveillance after therapy in both MIBC and NMIBC [42]. The role of NK cells is more prominent in the BCG therapy of NMIBC, as will be described below.

### 2.2. Adaptive Immune Effectors

T lymphocytes, especially their capacity for antigen-directed cytotoxicity, have become a central focus to engage the adaptive immune system in the fight against cancer [43]. The roles of T cells in BC immunity have remained largely unknown. Recent studies in humans and mice have revealed distinct activities of different T cell subtypes, which, for the most part, appear to be detrimental to the host, including the suboptimal clearance of bacteria or cancer cells, and/or boosting autoinflammation [44]. Several studies have characterized T cell responses in human and animal models of BC, with the goal of developing efficacious treatment strategies. A diverse range of T cell subtypes was identified, including pro-inflammatory cytotoxic CD8+ T cells (CTLs), anti-inflammatory Tregs, and CD4+ helper T cells (Th), especially those of the Th1 type [45], revealing notable T cell heterogeneity in bladder tumors [46]. Although their cytotoxic function is independent of HLA recognition, Tγδ lymphocytes also participate in immune surveillance against cancer by secreting pro-inflammatory cytokines such as INF-γ. T cells infiltrating solid tumors can play opposite roles, either inhibiting or promoting tumor growth [44]. Indeed, a single-cell RNA sequencing analysis of T cell populations in BC revealed notable T cell heterogeneity [46,47]. Although T cells are residents in the healthy human bladder, Th1 cells primed by DCs in the lymph node are recruited to the TME, where secreting IFN-γ can stimulate the antigen presentation of DCs via CD40/CD40L and the CTL function, as well as polarize macrophages to the M1 pro-inflammatory phenotype [46]. An analysis of the immune infiltrates in BC and lymph nodes after the checkpoint blockade in murine models revealed an expansion of IFN-γ producing Th1 cells; in addition, the neutralization of IFN-γ abolished the anti-tumor effect of the treatment, suggesting a key role for Th1 cells [48]. The Th1 biased response is also essential for a successful BCG therapy [49], as we will review below.

However, tumor antigen-specific T cells are susceptible to suppression by the TME. In fact, a higher CD4+ T cell density within the tumor was found to correlate with a poor prognosis in NMIBC [50,51]. Tumor-promoted Th2 polarization can suppress T cell anti-tumor response by secreting IL-4, IL-5, and IL-10; in fact, untreated or progressive BC is associated with lower levels of IFN-γ and IL-2 and higher levels of Th2 cytokines [52]. In addition, in response to factors secreted in the TME, Tregs are recruited from the circulation, further contributing to diminishing tumor immune surveillance by secreting IL-10 and TGF-β [53,54], or possibly by killing anti-tumor effectors and APCs [54]. In addition, it has also been proposed that the milieu of the TME might convert effector CD4+ T cells into Tregs or promote the differentiation of naïve CD4+ T cells into induced Tregs [55], further exacerbating the suppression of nascent anti-tumor immunity. This finding is in concordance with the fact that increased frequencies of Tregs are associated with poorer cancer patient prognosis [56].

After activation, T cells express the PD-1, which will interact with its ligand PD-L1 expressed on most tumor cells, suppressing anti-tumor activities of T cells by limiting their effector functions [57]. By far, the most important achievement in cancer treatment in recent decades has been the introduction of T cell-targeted immunomodulators that block the immune checkpoints [58]. Although the results are not entirely satisfactory, clinical trials and experimental models are underway to enhance anti-tumor T cell functions with monoclonal antibodies that block the interaction of PD-1 with PD-L1 [57,59] and other molecules that suppress T lymphocyte function, such as CTLA-4 [60] or tumor necrosis factor receptor 2 (TNFR2) [61], which will be reviewed in the following sections.

### 2.3. Tumor Immunoevasion

As mentioned in the preceding paragraphs, tumors promote the formation of highly immunosuppressive TMEs, preventing the generation of effective anti-tumor immune response through multiple mechanisms such as limiting T cell effector functions by engaging PD-1 and CTLA-4 [7,8]. In BCG treatment, the mechanisms of immune-escape are diverse, including the loss of MHC-I [62] or the up-regulation of PD-L1 and the poliovirus receptor (PVR or CD155) in tumor cells [63,64]. The expression of CD155 is associated with a poor prognosis and enhanced tumor progression in BC [65,66]. The specific blockade of the CD155 interaction with multiple inhibitory receptors expressed on NK and T lymphocytes, such as TIGIT, killer-cell immunoglobulin-like receptor (KIR) KIR2DL5 and CD96 [67], are now being explored in various types of cancer [10]. This multiplicity of inhibitory interactions between the tumor cell and the immune effectors is leading to numerous trials in which different combinations of multiple ICIs are being tested to prevent tumor immune escape.

Moreover, urothelial tumors secrete various immunosuppressive and anti-apoptotic factors including TGF-beta, PGE2, IL-10, and IL-6 [15,68,69,70], creating a highly tolerogenic TME, tightly linked to the accumulation of several types of immunosuppressive cells as MDSCs, tolerogenic DCs, TAMs, and T regs [69,70]. Thus, new cytokine therapies and diverse vaccines are being tested, sometimes in combination with various ICIs, to modulate the adverse TME and favor more effective antitumor responses.

## 3. Immunotherapy

Immunotherapy stimulates the patient’s own immune system to improve the recognition and attack of abnormal cells [71]. The detection and destruction of damaged or transformed cells by the immune system is a complex mechanism that develops as a result of the coordination of many cells and factors [72,73]. Immunotherapy is normally used to complement other treatments such as surgery, chemotherapy, and radiotherapy [74]. The efficacy of cancer immunotherapy has been demonstrated in in vitro and in vivo studies, as well as in clinical trials [71,75]. In fact, immunotherapy has revolutionized cancer treatment, including melanoma, renal, bladder, and lung cancers [33,71]. The most significant achievement in the last decade has undoubtedly been the introduction of T cell-targeted immunomodulators that block the immune checkpoints PD1/PDL1 and CTLA-4 [43,58]. However, many other cancer immunotherapies have been approved by the Food and Drug Administration (FDA) in the last thirty years, such as monoclonal antibody (mAb) for B-cell malignancies or engineered chimeric antigen receptor (CAR) T cells for hematological cancers. Immunotherapy can be used as a first line treatment [76] but also in combination with other treatments [77].

At present, the type of immunotherapy in BC is conditioned by the stage and the risk of the tumor. In high risk NMIBC, the therapy of choice is the intravesical treatment with BCG, while, in MIBC, it is the anti-checkpoint therapy (IT) in combination with chemo-radiotherapy.

### 3.1. BCG Immunotherapy in NMIBC

BCG is a live weakened form of *Mycobacterium bovis* used successfully for the first time in nine patients with BC in 1976 [13]. In 1980, Lamm et al. reported that BCG therapy after TURBT reduced the chance of relapse and progression [78,79]. BCG immunotherapy achieves a high percentage of positive responses, 55 to 65% for papillary tumors and 70 to 75% for CIS [33,80]. Although BCG therapy is currently the gold standard of care for high-risk NMIBC [81,82], 30 to 35% of patients will not respond to BCG therapy [16], so it is necessary to understand the immunobiology of BCG-induced tumor immunity in order to adapt more specific treatments for each patient to increase their efficacy and tolerability.

Although the specific mechanisms of BCG immunotherapy in the treatment of BC remains under investigation [16,72], it is well known that BCG induces a robust innate and adaptive immune response, triggering a cascade of events capable of modifying the TME and inducing a systemic immune modulation (Figure 1B). The binding and internalization of BCG to urothelial cells may occur through non-specific interactions or through fibronectin adhesion protein (FAP) [83], particularly to poorly differentiated cells [84], which may explain its greater efficacy in high-grade tumors. This internalization induces the secretion of IL-6, IL-8, GM-CSF, and tumor necrosis factor α (TNF-α), promoting the recruitment of immune cells to the bladder [84,85]. Although the internalization of BCG and its direct cytotoxic effect on cancer cells have been confirmed in vitro, its relevance in vivo has not been conclusively validated. Nonetheless, upon BCG internalization, BC cells increase nitric oxide (NO) production via inducible nitric oxide synthase (iNOS) [86], which may have a cytotoxic effect on urothelial cancer cells.

The activation of the innate immune response by BCG begins with the release of several pathogen-associated molecular patterns (PAMPs) that bind to pattern recognition receptors (PRRs) on APCs and macrophages, such as toll-like receptors TLR2, TLR4, and TLR9 [87] (Figure 1B). This interaction activates the MyD88 signaling pathway, which regulates the production of pro-inflammatory cytokines [88]. BCG immunotherapy recruits circulating macrophages and APCs to the bladder [89,90], which gets activated and produces IL-6, IL-12, and TNF-α, promoting polarization towards M1 macrophages and Th1. It has been described that M2 TAMs are associated with unfavorable response to BCG treatment, highlighting the importance of their differentiation with M1 macrophages for therapeutic success [91,92]. Furthermore, both macrophages and DCs bind to phagocyte BCG, which is crucial for processing and presenting BCG antigens to CD4 or CD8 T lymphocytes [93].

BCG treatment attracts T cells to the bladder mucosa, where they persist for several months and play a crucial antitumor activity [89,94]. The increase in CD4 T cells has been significantly correlated with a better response to BCG [89,94]. CD4 T lymphocytes can differentiate into several subtypes depending on the cytokine profile present in the environment. When exposed to IFN-γ, IL-12, or TNF-α, CD4+ T cells predominantly differentiate into Th1 [95], which produce pro-inflammatory cytokines and enhance the CTL activity. In contrast, in the presence of cytokines such as IL-4 or IL-10, CD4+ T lymphocytes differentiate into Th2 [49], which produces IL-4, IL-5, and IL-10, essential to induce the activation, proliferation, the differentiation of B cells, and the production of specific antibodies that help neutralize pathogens and mark target cells for destruction by other immune cells. A third major fate for CD4+ T lymphocytes is their differentiation into Th17 in the presence of IL-6 and TGF-β [96], which are known for their role in defending against fungi and extracellular bacteria, as well as mediating inflammatory responses. A recent study has demonstrated that BCG promotes an increase in the local levels of FLT3LG [97], which directly activates CTLs, thereby reinforcing the antitumor immune response. Once activated, CD8 T lymphocytes release IFN-γ, IL-2, and TNF-α, crucial cytokines for the differentiation of T cells into CTLs, which are responsible for the direct destruction of tumor cells (Figure 1B). Similarly, BCG exposure expands γδ T cells that show significant cytotoxic activity, suitable for tumor elimination [98].

Additionally, BCG promotes NK cells to produce IL-1β and IL-6 pro-inflammatory cytokines, contributing to the immune response [99]. NK cells have gained special relevance in the context of BCG treatment in BC. Although they do not constitute a predominant population in the healthy urothelium [100], these immune cells have demonstrated a capacity to respond to malignant cells in the bladder, making them a key component in the antitumor immune response. A notable expansion of CD56^high^ NK cell subpopulation has been observed after BCG therapy (Figure 1B), exhibiting functional maturity and antitumor cytotoxic effects [101]. These results reaffirm the essential role of NK cells in BC and open the door for further research to explore their therapeutic potential. In addition, the genetic profile of NK cell receptor interactions with host and tumor ligands can help to predict patient outcome and contribute to better treatment personalization [42].

Future treatments with the BCG cell wall skeleton (BCG-CWS), to replace live BCG, will be able to induce the same immunological response but without the risk of systemic infection. BCG-CWS nanoparticles administered intravesically in rodent models have shown to inhibit tumor growth [102]. More recently, nano-BCG and genetically engineered BCG have been proposed due to their ability to induce stronger and more stable immune responses [102]. The combined therapy of BCG and PD-1 inhibitors, such as pembrolizumab, for high-risk NMIBC (NCT02808143 and NCT02324582) may result in significant changes in the current practice for NMIBC [103,104].

### 3.2. Immune Checkpoint Inhibitors

Immune checkpoints are cellular receptors that, interacting with their ligands, function like regulatory “switches” of the immune system. Their primary role is to maintain immune balance by preventing immune cells from attacking healthy tissues, which could lead to autoimmune diseases [105]. These receptors are located on the surface of the main cytotoxic effectors, T lymphocytes, or NK cells (Figure 2). T lymphocyte activation requires the specific antigen recognition of peptides presented on the MHC, accompanied by co-stimulatory signals such as those delivered by CD28 and cytokines. Once activated, they proliferate and perform their effector functions, which must be subsequently stopped to return to homeostasis. Tumor cells can exploit these immune checkpoints to evade immune response by expressing ligands that interact with these inhibitory receptors on immune effectors [106]. These molecules are suitable targets for potentiating tumor immune responses. In the same way, NK cells require a balance between activating and inhibitory signals for their correct functioning. Blocking inhibitory signals, such as those promoted by NKG2A or TIGIT, may favor more effective immune responses (Figure 2).

ICIs represent a form of cancer immunotherapy designed to enhance the body’s natural immune response against cancer cells. These agents work by blocking the interactions of PD-1, CTLA-4, NKG2A, TIGIT, etc. with their ligands on cancer cells, releasing the “brakes” of T and NK cells to eliminate cancer cells more efficiently, promoting long-lasting anti-tumor responses (Figure 2). Responsiveness to checkpoint inhibitors is the key to successful cancer therapy, and ICI efficacy is affected by many factors, such as tumor genomic variability, host germline genetics, and the microbiome or PD-L1 expression levels [107]. Mutations on several signaling pathways may influence the effectiveness of ICIs [108], such as mutations in the Janus kinase (JAK) signaling pathway that are associated with enhanced apoptosis and the attenuated proliferation of T cells [109].

ICIs have demonstrated significant success in treating various types of cancer, including melanoma, lung cancer, renal cancer, and BC [110,111,112,113]. Summarized in Table 1, we described the clinical trials that have explored or are currently exploring the utility of ICIs in BC (Table 1).

#### 3.2.1. Anti-PD-1/PD-L1 Therapies

PD-1 is a membrane receptor found on the surface of various immune cells, including mature T lymphocytes, NK cells, B lymphocytes, and macrophages [123]. Upon activation, PD-1 binds to its ligands, PD-L1 and PD-L2, which are expressed on the surface of APCs and some tumor cells. This binding suppresses the activation and function of T lymphocytes, leading to a decrease in immune responses against self-antigens and tumor antigens. The PD-1/PD-L1 blockade lifts the inhibitory signals, allowing T lymphocytes to become activated and mount a more robust tumor immune response. PD-1 inhibitors include nivolumab and pembrolizumab, while PD-L1 inhibitors include atezolizumab, avelumab, and durvalumab (Figure 2).

##### Nivolumab

Nivolumab is a fully humanized IgG4 PD-1 antibody, which was approved in 2017 as a second-line treatment for platinum-resistant metastatic urothelial carcinoma (mUC), based on the results of the CheckMate 275 (NCT02387996) clinical trial [124]. This treatment demonstrated antitumor activity in patients with mUC who had previously undergone platinum-based therapy [125]. This phase II clinical trial enrolled 270 patients who received nivolumab at a dose of 3 mg/kg every two weeks. The findings indicated an overall response rate (ORR) of approximately 20% in patients treated with nivolumab, compared to 10% in the control group. Furthermore, response rates of 28.4%, 23.8%, and 16.1% were observed based on tumor cell PD-L1 expression >5%, >1%, and <1%, respectively. Patients with higher levels of PD-L1 also exhibited improved median OS, with 11.3 compared to 5.95 months in those with a lower PD-L1 expression. Nivolumab has an acceptable safety profile [124]. Follow-up in 2020 revealed that nivolumab continues to deliver durable antitumor activity in patients with platinum-resistant mUC [126]. However, it is important to note that a subset of patients experienced treatment-related adverse effects, including three fatalities: one case of acute respiratory failure, pneumonitis, and/or cardiac compromise. These findings underscore the necessity of ongoing monitoring and the careful management of patients receiving nivolumab.

The CheckMate-274 trial is a pivotal phase III study evaluating the efficacy of nivolumab as adjuvant therapy in 709 patients with high-risk MIBC following radical surgery. Results demonstrated a median disease-free survival (DFS) of 20.8 months for patients treated with nivolumab, compared to 10.8 months for the placebo group, with a hazard ratio of 0.70 (*p* < 0.001). In the subpopulation with a PD-L1 expression ≥1%, 74.5% of patients receiving nivolumab were alive and disease-free at six months, versus 55.7% in the placebo group. These findings underscore the potential of nivolumab to improve outcomes in high-risk MIBC patients and suggest its integration into clinical practice. This study will be ongoing till 2027 (but not recruiting patients), when progression-free survival will be analyzed over time (NCT02632409) [127].

##### Pembrolizumab

Pembrolizumab is a humanized anti-PD-1 antibody that blocks PD-1 interaction with both PD-L1 and PD-L2. The treatment has been shown to be beneficial across multiple stages of the disease. The FDA approved this drug as a second-line treatment in patients with mUC because of the positive survival data from KEYNOTE-045 (NCT02256436), which demonstrated higher median OS for patients treated with pembrolizumab than patients treated with chemotherapy. In fact, after five years, the benefits were maintained, including a durable response (median > 2 years) [128,129]. This phase III trial has already been completed and positively evaluated.

The KEYNOTE-052 (NCT02335424) phase II trial investigated the efficacy of pembrolizumab in 370 patients with advanced BC who were ineligible for cisplatin-based chemotherapy. This study, which has already been completed, obtained an ORR of 47% in patients with a higher expression of PD-L1 compared to 21% in patients with a lower expression of the ligand [130]. These results support the use of pembrolizumab as a first-line treatment for cisplatin-ineligible patients with locally advanced and unresectable or metastatic BC (mBC).

With nearly 5 years of follow-up, pembrolizumab monotherapy demonstrated durable efficacy in patients with platinum-resistant mUC in the KEYNOTE-045 trial and as first-line therapy in cisplatin-ineligible patients in the KEYNOTE-052 trial. Importantly, there were no new safety signals observed during this extended follow-up period, reinforcing the role of pembrolizumab as a viable treatment option for these patients [131].

The KEYNOTE-361 trial (NCT02853305) was a phase III study that compared the efficacy of pembrolizumab administered with and without chemotherapy against chemotherapy alone in patients with mBC. However, the study did not meet the predetermined efficacy thresholds necessary for statistical significance [116]. The PLUMMB clinical trial (NCT02560636) was designed to investigate the safety and efficacy of combining radiation therapy and pembrolizumab in patients with MIBC or locally advanced BC. The study initially used a dose of 36 Gy in six fractions and 100 mg of pembrolizumab, but it was suspended after five out of five patients experienced severe toxicities in the first cohort [132]. However, the trial was reopened in June 2020 with a dose of 24 Gy in four fractions and 100–200 mg of pembrolizumab but was suspended again after a severe toxicity. As a result, the radiation dose was reduced to 24 Gy in six fractions. Therefore, the authors concluded that it was not possible to safely combine 24 Gy in four fractions with 200 mg of pembrolizumab and recommended using a dose of 24 Gy in six fractions in the dose escalation phase of the study [133].

##### Atezolizumab

Atezolizumab is an anti- PD-L1 antibody, approved by the FDA for the treatment of patients with locally advanced or metastatic BC who has previously received chemotherapy containing platinum. Clinical trials, such as the IMvigor210 (NCT02108652) demonstrated that atezolizumab is associated with durable responses and improved survival rates in patients with BC. The study showed a significant ORR and a subset of patients achieving complete response, indicating the potential for the long-term benefits of this treatment. Atezolizumab has been utilized in various settings, including first-line treatment for patients who were not eligible for cisplatin-based chemotherapy, as well as a second line after chemotherapy failure like in the IMvigor130 trial (NCT02807636). Its effectiveness across different stages of BC underscores its versatility as a treatment option. The safety profile of atezolizumab is generally favorable compared to traditional chemotherapy. While immune-related adverse events can occur, they are often manageable, and many patients tolerate the treatment well. This aspect is particularly important for patients who may be frail or have comorbidities that complicate their treatment options. The IMvigor130 trial demonstrated that atezolizumab is a viable treatment option for patients with untreated locally advanced BC or mUC, both as a monotherapy and in combination with chemotherapy. The results support the integration of atezolizumab into the treatment paradigm for this patient population, highlighting its potential to improve OS and response rates while maintaining a manageable safety profile.

##### Avelumab

Avelumab is an IgG1 anti-PD-L1 antibody that inhibits PD-1 interaction with PD-L1, but not that with PD-L2. This treatment was evaluated by the JAVELIN solid tumor trial in 2017. A median OS of 13.7 months together with an ORR of 18.2% were initially reported by this study. Unfortunately, all 44 patients who participated in this trial developed adverse events. However, there was a trend toward a higher survival after 12 weeks of treatment in patients with a high expression of PD-L1 versus patients with PD-L1 negative tumors (ORR of 53.8% vs. 4.2%, respectively). The FDA approved Avelumab as a second line treatment for locally advanced or metastatic BC in platinum-refractory patients. Avelumab showed durable clinical activity and had a manageable and tolerable safety profile irrespective of PD-L1 expression [134].

JAVELIN bladder 100 (NCT02603432) is an open-label, phase III trial which investigated avelumab as a first-line maintenance therapy in patients with locally advanced or metastatic BC that had not progressed with chemotherapy (platinum) [135]. In 2020, the FDA approved avelumab as a treatment for these patients [135]. In addition, this trial was evaluated in a subgroup of Japanese patients, showing that first-line maintenance with avelumab represents a new standard of care in Japanese patients with advanced UC who have not progressed with first-line chemotherapy [136]. The longer-term follow-up continues to demonstrate clinically meaningful efficacy benefits for avelumab as a first-line maintenance therapy in combination with BSC (best supportive care) compared to BSC alone for patients with advanced UC [120]. Avelumab is a valid maintenance treatment option for patients responding to chemotherapy.

A phase IV trial, JAVELIN BLADDER 101, assessed the efficacy of avelumab in combination with chemotherapy in previously untreated patients with mBC. Preliminary results indicated that the combination of avelumab with chemotherapy could enhance response rates compared to chemotherapy alone, suggesting a synergistic effect.

The GCISAVE trial (NCT03324282) is a non-comparative randomized Phase II clinical study designed to evaluate the efficacy and safety of the combination of gemcitabine and cisplatin (GCis) with or without avelumab in the first-line treatment of patients with locally advanced or metastatic BC. Although the study was stopped prematurely due to the approval of avelumab as a maintenance treatment, indicating a shift in treatment protocols, they demonstrated the potential benefits of combining avelumab with gemcitabine and cisplatin in these cancers [137].

Avelumab has been extensively evaluated in combination with other types of immunotherapies, such as trial NCT03892642. The primary objective was to evaluate the efficacy and safety of avelumab combined with BCG in patients with high-risk NMIBC who have either relapsed or failed to respond adequately to standard BCG therapy. The combination of avelumab with BCG has the potential to improve outcomes for these patients.

In patients who have a worse progression of the disease and advanced UC, the combination of avelumab with radiation is being studied (NCT03747419). The study will also evaluate additional outcomes, including participants’ overall health status and quality of life during and after treatment. Currently in the recruitment phase, the trial is expected to conclude in 2030, allowing for an extensive evaluation of the treatment’s safety and efficacy.

##### Durvalumab

Durvalumab is a human IgG1 anti-PD-L1 antibody that blocks PD-1/PD-L1 interaction [138,139]. Durvalumab has been investigated in combination with other immunotherapeutic agents. The NIAGARA trial (NCT03732677) investigates the combination of durvalumab with chemotherapy in patients with MIBC, aiming to enhance pathological response and long-term survival. Meanwhile, the POTOMAC trial (NCT03528694) assesses the efficacy of combining durvalumab with BCG therapy in high-risk, BCG-naïve NMIBC patients, seeking to address the limitations of standard BCG treatment. Lastly, the NILE trial (NCT03682068) explores the use of durvalumab and tremelimumab in conjunction with standard chemotherapy for patients with unresectable locally advanced or metastatic UC, with the hope of improving clinical outcomes compared to chemotherapy alone. Collectively, these studies hold the potential to transform treatment approaches in these oncological conditions and may offer new therapeutic options for patients in the future. The NIAGARA, POTOMAC, and NILE clinical trials are currently ongoing and under evaluation, yet they appear promising in the treatment of various forms of BC.

#### 3.2.2. Anti-CTLA-4 Therapies

CTLA-4 is an immune checkpoint receptor that plays a crucial role in regulating the immune response. It is mainly expressed on T lymphocytes, where it acts as a negative regulator of their activation. CTLA-4 competes with CD28 [140,141] for binding to CD80 (also known as B7.1) and CD86 (also known as B7.2) on APCs [142]. CTLA-4 has a higher affinity for these ligands than CD28 [143], so it can deliver inhibitory signals. In the context of cancer, blocking CTLA-4 with ipilimumab or tremelimumab enhances the activity of effector T cells, which can lead to a more robust anti-tumor response [144] (Figure 2).

##### Ipilimumab

Ipilimumab was the first ICI approved by the FDA on 2011 for the treatment of stage III and IV melanoma [145]. This approval marked a milestone in cancer therapy, as it was the first immunotherapeutic treatment to demonstrate a long-term survival benefit in patients with melanoma [63].

Some trials have evaluated the combination of ipilimumab with nivolumab to treat patients with BC. The CheckMate 032 (NCT01928394) trial studied the safety and efficacy of the combination of ipilimumab and nivolumab in patients with advanced or metastatic solid tumors, including UC. The trial compared the effects of combined treatment versus each agent alone and found that the combination therapy demonstrated promising antitumor activity with an acceptable safety profile. Specifically, the NIVO3 (3 mg/kg) cohort achieved a 25.6% objective response rate, while the combination regimen of NIVO1 (1 mg/kg) + IPI3 (3 mg/kg) had the highest response rate (38.0%). The median duration of response exceeded 22 months across all treatment arms, highlighting the significant and durable benefits of these therapies [146].

The CheckMate 901 (NCT03036098) trial was designed to assess the efficacy and safety of the combination of nivolumab and ipilimumab in patients with MIBC who were undergoing neoadjuvant therapy prior to radical cystectomy. Results indicated that the combination therapy led to significant pathologic responses, with a proportion of patients achieving complete pathological response at the time of surgery. The trial’s findings suggested that the combination of nivolumab and ipilimumab could improve outcomes compared to traditional neoadjuvant chemotherapy [147].

##### Tremelimumab

Tremelimumab has been studied in other contexts, but its use in BC has not been well characterized. The DANUBE (NCT02516241) was a pivotal phase III trial designed to evaluate the effectiveness and safety of durvalumab alone or in combination with tremelimumab, compared to standard chemotherapy in patients with stage IV untreated, unresectable, locally advanced, or metastatic UC [148]. The study did not meet its co-primary endpoints of improving OS compared to standard chemotherapy for either durvalumab monotherapy or the combination of durvalumab and tremelimumab. Although the combination regimen showed a trend towards improved OS, it did not reach statistical significance [122]. Another clinical trial (NCT0281242) investigates the efficacy of two ICI, durvalumab and tremelimumab, in patients with high-risk MIBC who are ineligible for cisplatin-based therapy prior to surgery. So far, the data indicate that two cycles of neoadjuvant combination therapy have shown a tolerable safety profile and encouraging efficacy results [149]. These findings justify the need for future clinical trials to further develop the combination treatment with an anti-CTLA-4 and an anti-PD-L1 in patients with MIBC who are not eligible for cisplatin. Additionally, it is crucial to determine the optimal duration of this therapy to maximize benefits for this patient population. However, final results are still pending.

#### 3.2.3. Targeting NK and T Cell Immune Checkpoints

NK cytotoxic activity is highly regulated for different activating and inhibitory receptors, many of them shared with T lymphocytes. Among them, NKG2A, TIGIT, and KIR2DL5 have gained relevance in the last decade.

The interaction between NKG2A and HLA-E represents an alternative immune checkpoint axis that may be crucial for modulating immune responses in BC. NKG2A, predominantly expressed on NK and CD8+ T cells, can interact with its ligand HLA-E, often overexpressed in tumor cells [150], resulting in the inhibition of these cytotoxic effectors. Recent studies have indicated that blocking NKG2A can partially restore the functionality of NK and CD8+ T cells in an HLA-E-dependent manner [9,151]. This finding presents a compelling rationale for developing clinical trials that combine the NKG2A blockade with other established ICIs, particularly in tumors exhibiting elevated levels of HLA-E.

TIGIT is a novel immune checkpoint receptor with an inhibitory function, expressed on NK and T cells that interact with CD155 (PVR) and CD112 (PVRL2, nectin-2). TIGIT inhibition via novel monoclonal antibodies (Vibostolimab, Etigilimab, Sacituzumab, and Tiragolumab) represents an interesting therapeutic strategy to be exploited in early-phase clinical trials [10,152].

KIR2DL5 (CD158f), the most recently identified functional killer-cell immunoglobulin-like receptor, is expressed in NK and T lymphocytes and has gained interest in cancer immunotherapy with the identification of CD155 as its ligand [153], which is highly expressed in MIBC [154]. KIR2DL5 functions as an inhibitory receptor by binding to CD155 on tumor cells, which promotes the formation of inhibitory synapses and suppresses NK cell cytotoxicity [155]. CD155 expression has been associated with a higher risk of recurrence in NMIBC. Blocking the KIR2DL5/CD155 interaction with mAbs leads to enhance NK cell-mediated cytotoxicity against CD155+ tumors and reduced tumor growth in humanized tumor models [155].

##### Monalizumab

Monalizumab is a first-in-class IgG4 antibody that targets NKG2A. The COAST trial (NCT03822351) assesses the efficacy of monalizumab in combination with durvalumab in non-small-cell lung cancer [156]. Other groups have demonstrated beneficial effects of monalizumab on survival rates in pre-clinical mouse models of lung cancer and lymphoma. Monalizumab is currently undergoing various clinical trials to assess its efficacy in combination with ibrutinib for chronic lymphocytic leukemia (NCT02557516), trastuzumab for breast cancer (NCT04307329), durvalumab for colorectal cancer (NCT02671435), and cetuximab for head and neck squamous cell carcinoma (HNSCC) (NCT02643550).

In BC, the expression of NKG2A is associated with a better response to anti-PD-L1 therapy in cancers with higher levels of CD8A, PDCD1, or PD-L1, suggesting that NKG2A could be a predictive biomarker [9]. Altogether, these results support the idea that monalizumab could enhance anti-tumor responses and improve patient outcomes across multiple cancer types. The phase II clinical trial ENHANCE (Elevated NKG2A and HLA-E Amplify NK/CD8 Checkpoint Engagers, NCT06503614), not yet recruiting, will evaluate durvalumab (MEDI4736) plus monalizumab in NMIBC [157].

##### Tiragolumab and Sacituzumab

Tiragolumab and Sacituzumab are anti-TIGIT mAbs to block its interaction with CD112/CD155 ligands. Currently, there are clinical trials such as NCT05394337 (a phase I) to evaluate their safety in combination with neoadjuvant PD-1 (Atezolizumab) plus TIGIT (Tiragolumab) in patients with cisplatin-ineligible operable high-risk urothelial carcinoma and NCT05327530 (a phase II) to study the safety and efficacy of various combinations of avelumab (anti-PD-L1) with other targets such as TIGIT (Sacituzumab) as therapy in locally advanced or metastatic urothelial carcinoma (javelin bladder medley).

Nonetheless, two large phase III trials with TIGIT blockades recently failed to improve cancer outcomes. Recent studies have described that KIR2DL5, which share the ligand with TIGIT, can provide alternative inhibitions of NK and T cell functions, so that the used of combined therapies to block both inhibitory receptors would be more appropriate [155].

### 3.3. New Immunotherapies in the Pipeline

Recent advancements in BC immunotherapy have expanded beyond traditional treatments like BCG and ICIs. While BCG remains the standard for high-risk NMIBC, other modalities are showing promising results. Cytokine therapy, including interleukin-2 (IL-2) and interferon-alpha, has been explored to stimulate immune responses, though its use has been limited due to significant side effects. Ongoing research aims to optimize these treatments by minimizing toxicity and enhancing their effectiveness when combined with other therapies [158]. Vaccine therapies, such as the Ty21a vaccine, are gaining attention for their potential in NMIBC, showing significant antitumor responses and offering alternatives for patients who cannot tolerate BCG [159]. Dendritic cell vaccines are also under investigation to activate T cells against tumor-specific antigens, with some studies exploring their combination with other immunotherapies [160,161].

Antibodies-drug conjugates (ADCs) have emerged as a new therapeutic option in treating BC, especially in advanced stages or when the tumor becomes refractory to standard treatments. Enfortumab vedotin, a mAb targeting nectin-4, has shown substantial efficacy to prolong survival for patients with advanced urothelial carcinoma who previously received chemotherapy and PD-1/PD-L1 inhibitors [162,163].

Oncolytic adenoviral therapy plus pembrolizumab in BCG-unresponsive NMIBC are under investigation in the phase 2 CORE-001 trial [164]. Cretostimogene grenadenorepvec is a serotype-5 oncolytic adenovirus designed to selectively replicate in cancer cells with retinoblastoma pathway alterations, previously tested as monotherapy in BCG-treated NMIBC. In this trial, the potential synergistic efficacy between intravesical cretostimogene and systemic pembrolizumab in patients with BCG-unresponsive NMIBC with CIS was assessed. This combined treatment demonstrated enduring efficacy, with a toxicity profile similar to its monotherapy components; therefore, this combination may shift the benefit-to-risk ratio for patients with BCG-unresponsive CIS.

## 4. Concluding Remarks

As new co-inhibitory and co-stimulatory receptors have been discovered and their functions investigated over the past decades, new immunotherapeutic treatments have been successfully tested in experimental models and clinical trials. Although this research has been a major breakthrough in the treatment of advanced cancer, most patients do not gain significant benefit. The introduction of new therapeutic targets and their combination with existing ones should constitute the new frontier in advanced BC treatment research, with the aim of achieving responses in the majority of patients.

## Figures and Tables

**Figure 1 cells-13-01937-f001:**
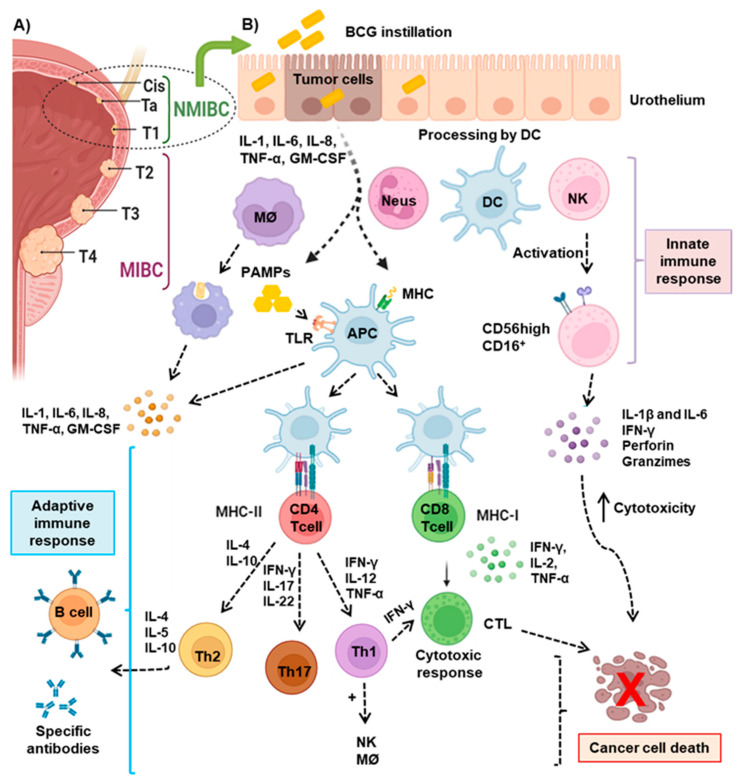
Immunological mechanisms of intravesical BCG treatment in bladder cancer: (**A**) Classification of BC according to the TNM staging system. CIS (carcinoma in situ) represents a non-muscle invasive bladder cancer (NMIBC) confined to the bladder lining; Ta, papillary NMIBC limited to the inner lining; T1, NMIBC that invades the subepithelial connective tissue without penetrating the muscle layer; T2, muscle-invasive bladder (MIBC) cancer; T3, MIBC that invades the perivesical tissue surrounding the bladder; and T4, advanced MIBC that invades surrounding structures such as the prostate, uterus, or pelvic wall. (**B**) NMIBC is treated with BCG. Upon instillation, BCG is taken up by bladder urothelial cells, antigen-presenting cells (APC), macrophages, and dendritic cells (DCs), leading to the release of pro-inflammatory cytokines and the activation of the immune response, including T and natural killer (NK) cells, which recognize and attack tumor cells. DCs express toll-like receptors (TLRs) that recognize pathogen-associated molecular patterns (PAMPs), promoting the secretion of cytokines and the presentation of tumor antigens via the major histocompatibility complex (MHC) to CD4+ and CD8+ T lymphocytes, thus contributing to tumor eradication. BCG induces NK cell functional maturation, increasing the expression of CD56 and the release of proinflamatory cytokines, granzyme, and perforin, which contribute to the destruction of tumor cells. Understanding these mechanisms is vital for optimizing BCG therapy and improving outcomes for patients with bladder cancer.

**Figure 2 cells-13-01937-f002:**
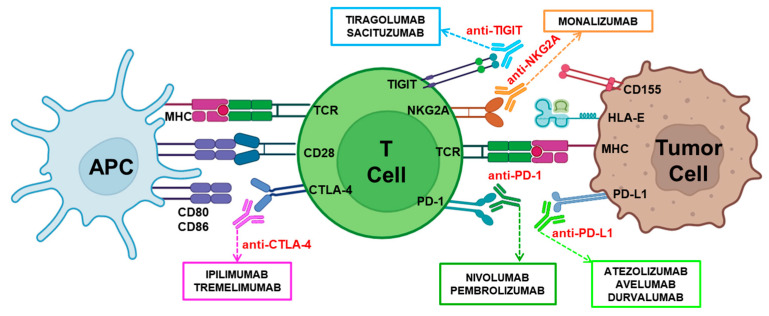
Mechanisms of immune checkpoint blockade in cancer therapy. The adequate activation of T lymphocytes requires a primary specific signal delivered by the TCR/MHC interaction together with co-stimulatory signals mainly delivered by the CD28/CD80-CD86 interaction. In contrast, the interactions of CTLA-4/CD80-CD86, PD-1/PD-L1, NKG2A/HLA-E, or TIGIT/CD155 inhibit and regulate T cell activation and function. These inhibitory interactions can be blocked using immunotherapeutic monoclonal antibodies: anti-PD-1 (Nivolumab, Pembrolizumab), anti-PD-L1 (Atezolizumab, Avelumab, and Durvalumab), anti-CTLA-4 (Ipilimumab, Tremelimumab), anti-NKG2A (Monalizumab), or anti-TIGIT (Tiragolumab, Sacituzumab).

**Table 1 cells-13-01937-t001:** Ongoing clinical trials of checkpoint inhibitors in bladder cancer.

Trial (Year)	NCI Identifier	Phase	Line of Treatment	Estimated N	Treatment	Status	Projected End	OS (Months)	PFS (Months)	Source
IMvigor210 * (2016/17)	NCT02108652	II	Locally advanced or MIBC	310	Atezolizumab	Approved	February 2023	7.9	NA	[114]
IMvigor130 * (2020)	NCT02807636	III	UC	1213	Platinum-based chemotherapy (A) vs. Atezolizumab + platinum-based chemo (B) vs. atezolizumab monotherapy (C)	Active, not recruiting	December 2024	13.44 vs. 16.13 vs. 15.21	6.34 vs. 8.18	ClinicalTrials.gov
Keynote-045 * (2017)	NCT02256436	III	Advanced UC	542	Chemotherapy vs. Pembrolizumab	Completed	August 2021	7.4 vs. 10.3	2.1 vs. 3.3	ClinicalTrials.gov
KEYNOTE-052 *	NCT02335424	II	Advanced/unresectable or metastatic UC who are ineligible for cisplatin-based therapy	374	Pembrolizumab	Completed	February 2022	11.3	2.2	[115]
KEYNOTE-361 * (2016)	NCT02853305	III	UC	1010	Pembrolizumab alone vs. Pembrolizumab + chemorherapy vs. chemotherapy	Completed	September 2022	15.6 vs. 17 vs. 14.3	8.3 vs. 7.1	[116]
CheckMate-275 * (2017)	NCT02387996	II	Metastatic or unresectable bladder cancer	270	Nivolumab	Completed	November 2021	8.6	1.9	[117]
CheckMate-274 * (2021)	NCT02632409	III	High-risk MIBC	709	Nivolumab vs. Placebo	Active, not recruiting	May 2027	69.5 vs. 50.1	39.4 vs. NR	[118]
JAVELIN Solid Tumor (2017)	NCT01772004	I	Metastatic or locally advanced solid tumors including BC	1756	Avelumab	Completed	December 2019	10.7	4.1	[119]
JAVELIN BLADDER 100 (2020)	NCT02603432	III	Locally advanced or MIBC	700	Avelumab maintenance + BSC vs. BSC	Active, not recruiting	March 2023	23.8 vs. 15.0	5.5 vs. 2.1	[120]
Study 1108 * (2017)	NCT01693562	I/II	Advanced solid tumors including UC	1022	Durvalumab	Completed	February 2020	23.8	5.9	[121]
NIAGARA	NCT03732677	III	MIBC	1063	Durvalumab + gemcitabine/cisplatin (neoadjuvant) vs. durvalumab (adjuvant)	Active, not recruiting	June 2026	NA	NA	
-	NCT02812420	Early I	High-risk BC ineligible for neoadjuvant cisplatin chemotherapies	54	Durvalumab+ tremelimumab	Active, not recruiting	December 2024	NA	NA	
POTOMAC (2018)	NCT03528694	III	High-risk, BCG-naïve NMIBC	1019	Durvalumab +BCG vs. BCG	Active, not recruiting	November 2024	NA	NA	
DANUBE * (2015/20)	NCT02516241	III	Stage IV UC	1126	Durvalumab+ Tremelimumab vs. Durvalumab vs. Chemotherapy	Active, not recruiting	December 2024	14.4 vs. 15.1 vs. 12.1	NA	[122]
Checkmate 032 (2016)	NCT01928394	I/II	Advanced or metastatic solid tumors (BC)	1163	Nivolumab monotherapy vs. nivolumab + ipilimumab	Active, not recruiting	October 2024	NA	NA	
CheckMate901 (2017)	NCT03036098	III	Untreated inoperable or metastatic UC	1290	Nivolumab + ipilimumab vs. nivolumab + chemotherapy vs. chemotherapy	Active, not recruiting	June 2028	NA	NA	
NILE (2018)	NCT03682068	III	Unresectable locally advanced or metastatic UC	1246	Durvalumab + chemotherapy vs. durvalumab + tremelimumab + chemotherapy vs. chemotherapy alone	Active, not recruiting	June 2025	NA	NA	
PLUMMB (2016)	NCT02560636	I	MIBC	34	Pembrolizumab + Radiotherapy	Unknown Status	June 2024	NA	NA	
KEYNOTE-676 (2018)	NCT03711032	III	High-risk NMIBC persistent or recurrent after induction BCG or BCG-naïve	1525	Pembrolizumab + BCG	Recruiting	November 2024	NA	NA	
ALBAN	NCT03799835	III	High-risk, BCG-naïve NMIBC	516	Atezolizumab + BCG vs. BCG	Recruiting	February 2028	NA	NA	
ENHANCE	NCT06503614	II	NMIBC	60	Durvalumab + Monalizumab	Not yet recruiting	December 2026	NA	NA	
-	NCT05394337	I	High-risk UC who are ineligible for cisplatin before surgery	10	Atezolizumab + Tiragolumab	Recruiting	January 2026	NA	NA	
JAVELIN Bladder Medley	NCT05327530	II	Locally advanced or metastatic UC	256	Avelumab Avelumab + Sacituzumab govitecan Avelumab + M6223 Avelumab + NKTR-255	Active, not recruiting	January 2025	NA	NA	

* PD-L1 was a predictive biomarker in these trials. OS: overall survival; PFS: progression-free survival; NA: not available, NR: not reached.

## Data Availability

The data of the work will be available upon justified request of the researchers who need it.

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
