# Peer review of "Exploring the Immunoresponse in Bladder Cancer Immunotherapy"

_cells, 2024, doi:10.3390/cells13231937_

Round 1
Reviewer 1 Report
Comments and Suggestions for Authors
Bladder cancer (BC) is a highly immunogenic disease that involves both innate and adaptive immune components, however, little is still known about how immune cells respond to BC. Immunotherapy with BCG remains the gold standard treatment for high-risk NMIBC, while the use of immune checkpoint inhibitors (anti-CTLA4 and anti-PD1/PD-L1) to boost natural immune vigilance is emerging as an effective therapy for invasive and metastatic urologic tumors. This review aims to unravel the immune responses in bladder cancer, which will contribute to establishing new and promising therapeutic options.
1. Authors should more comprehensively summarize the latest progress in bladder cancer immunotherapy, including other immunotherapy methods besides immune checkpoint inhibitors, such as cytokine therapy and vaccine therapy. According to “Intravesical Ty21a treatment of non-muscle invasive bladder cancer induces immune responses that correlate with safety and may be associated to therapy potential. J Immunother Cancer. 2023.”, they also provided a novel vaccine therapy compared with BCG. Besides, authors should update more latest references.
2. The mechanism of bladder cancer immunotherapy should be more in-depth discussed, such as the specific roles of immune cells and immune evasion mechanisms, in order to better guide clinical treatment.
3. The results of clinical trials on bladder cancer immunotherapy should be more systematically analysized, including the comparison of efficacy of different treatment regimens and the identification of predictive biomarkers, to provide more evidence for clinical practice.
4. This article should provide more suggestions on the future development direction of bladder cancer immunotherapy, such as combination therapy strategies and the development of new targets, to provide ideas for further improving the treatment outcomes of bladder cancer patients.
5. The structure of the article should be more reasonable, such as providing a comprehensive overview of the research background on bladder cancer immunotherapy in the introduction, and offering more suggestions on future development directions in the discussion section.
6. Authors should pay more attention to the consistency of the format, such as the requirements for title hierarchy, paragraph indentation, line spacing, etc.
Comments on the Quality of English LanguageOverall, the English expression level of this paper is relatively high, with smooth sentences and strong logic. Authors have maintained a good balance between general summary and specific analysis. However, some sentences are slightly complex, and the sentence structure can be appropriately simplified to improve readability and transitional sentences between paragraphs can be further optimised to make the entire article more fluent.
Author Response
Comment-1: Bladder cancer (BC) is a highly immunogenic disease that involves both innate and adaptive immune components, however, little is still known about how immune cells respond to BC. Immunotherapy with BCG remains the gold standard treatment for high-risk NMIBC, while the use of immune checkpoint inhibitors (anti-CTLA4 and anti-PD1/PD-L1) to boost natural immune vigilance is emerging as an effective therapy for invasive and metastatic urologic tumors. This review aims to unravel the immune responses in bladder cancer, which will contribute to establishing new and promising therapeutic options.
- Authors should more comprehensively summarize the latest progress in bladder cancer immunotherapy, including other immunotherapy methods besides immune checkpoint inhibitors, such as cytokine therapy and vaccine therapy. According to “Intravesical Ty21a treatment of non-muscle invasive bladder cancer induces immune responses that correlate with safety and may be associated to therapy potential. J Immunother Cancer. 2023.”, they also provided a novel vaccine therapy compared with BCG. Besides, authors should update more latest references.
Response-1. New immunotherapeutic treatments suggested by the reviewer have been briefly reviewed. We have also included new lines of immunotherapies targeting simultaneous inhibitory receptor of T and NK cells. Thank you very much for your suggestions.
Comment-2. The mechanism of bladder cancer immunotherapy should be more in-depth discussed, such as the specific roles of immune cells and immune evasion mechanisms, in order to better guide clinical treatment.
Response-2. New section (2.3) has been added to highlight the immune evasion mechanisms that compromise the tumor immune surveillance.
Comment-3 The results of clinical trials on bladder cancer immunotherapy should be more systematically analysized, including the comparison of efficacy of different treatment regimens and the identification of predictive biomarkers, to provide more evidence for clinical practice.
Response-3. The table has been modified following the reviewer's suggestions, identifying the predictive markers and the efficacy of the trials reflected as progression-free and overall survival. The objective response rate (ORR) has not been included since this information was available in very few trials.
Comment-4 This article should provide more suggestions on the future development direction of bladder cancer immunotherapy, such as combination therapy strategies and the development of new targets, to provide ideas for further improving the treatment outcomes of bladder cancer patients.
Response-4. These questions have been addressed in a new section (3.3) at the end of the manuscript.
Comment-5 The structure of the article should be more reasonable, such as providing a comprehensive overview of the research background on bladder cancer immunotherapy in the introduction, and offering more suggestions on future development directions in the discussion section.
Response-5. These suggestions have been addressed in new paragraphs along the introduction and the rest of the manuscript.
Comment-6. Authors should pay more attention to the consistency of the format, such as the requirements for title hierarchy, paragraph indentation, line spacing, etc.
Response-6. We have edited the format, to make clearer the different sections of the manuscript.
Comment-7. Comments on the Quality of English Language Overall, the English expression level of this paper is relatively high, with smooth sentences and strong logic. Authors have maintained a good balance between general summary and specific analysis. However, some sentences are slightly complex, and the sentence structure can be appropriately simplified to improve readability and transitional sentences between paragraphs can be further optimised to make the entire article more fluent.
Response-7. We have given a careful reading to the manuscript trying to include new punctuation marks, to make text more readable.
Reviewer 2 Report
Comments and Suggestions for Authors
The authors aim to review immune responses in bladder cancer, standard therapy (surgery, chemo, and immune therapy), and ongoing clinical studies with immunotherapeutics. When detected, most tumours are in an early stage and require local (intravesical) therapy (non-muscle-invasive bladder cancer). In contrast, muscle-invasive and metastatic bladder cancer and primary urethral carcinoma need another type of therapy.
Section 2 deals with innate and adaptive immune effectors, and section 3 deals with immunotherapy for all types of bladder cancer.
Exhaustive descriptions are provided of immune cell types, factors released by them, either stimulating or suppressing immune system activity (section 2) and clinical studies with immunotherapeutics, predominantly immune checkpoint inhibitors (section 3). Although this is commendable, it makes reading and understanding the text very difficult. Perhaps a solution could be a combination of text focusing on the main issues and details in figures (description in the legends).
Figures 1 and 2 and Table 1 are excellent.
Minor remarks:
Line 65-unfortunately: delete this word because tumours always aim to survive by suppressing the immune system.
Conflicts of Interest: The authors declare conflicts of interest. Is this correct, or is it a typing error?
Author Response
The authors aim to review immune responses in bladder cancer, standard therapy (surgery, chemo, and immune therapy), and ongoing clinical studies with immunotherapeutics. When detected, most tumours are in an early stage and require local (intravesical) therapy (non-muscle-invasive bladder cancer). In contrast, muscle-invasive and metastatic bladder cancer and primary urethral carcinoma need another type of therapy.
Section 2 deals with innate and adaptive immune effectors, and section 3 deals with immunotherapy for all types of bladder cancer.
Comment-1. Exhaustive descriptions are provided of immune cell types, factors released by them, either stimulating or suppressing immune system activity (section 2) and clinical studies with immunotherapeutics, predominantly immune checkpoint inhibitors (section 3). Although this is commendable, it makes reading and understanding the text very difficult. Perhaps a solution could be a combination of text focusing on the main issues and details in figures (description in the legends).
Response-1. The manuscript has been carefully reread to add new information and complement the previous data, making new calls to the figures to better illustrate what is described in the manuscript.
Comment-2. Figures 1 and 2 and Table 1 are excellent.
Response-2. Thanks.
Minor remarks:
Comment-3. Line 65-unfortunately: delete this word because tumours always aim to survive by suppressing the immune system.
Response-3. We have changed “Unfortunately” for “Nonetheless”
Comment-4. Conflicts of Interest: The authors declare conflicts of interest. Is this correct, or is it a typing error?
Response-4. No. We do not have any conflict of interest.